# Learning Dense Visual Descriptors using Image Augmentations for Robot Manipulation Tasks

**Christian Graf[†,1], David B. Adrian[†,1,2], Joshua Weil[1,3], Miroslav Gabriel[1],**

**Philipp Schillinger[1], Markus Spies[1], Heiko Neumann[2], Andras Kupcsik[1]**

[†]Equal contribution, [1]Bosch Center for Artifical Intelligence,
[2]Ulm University, [3]KTH Royal Institute of Technology

**Abstract:** We propose a self-supervised training approach for learning view-invariant dense visual descriptors using image augmentations. Unlike existing works, which often require complex datasets, such as registered RGBD sequences, we train on an unordered set of RGB images. This allows for learning from a single camera view, e.g., in an existing robotic cell with a fix-mounted camera. We create synthetic views and dense pixel correspondences using data augmentations. We find our approach to be competitive compared to existing methods, despite the simpler data recording and setup requirements. We show that training on synthetic correspondences provides descriptor consistency across a broad range of camera views. We compare against training with geometric correspondence from multiple views and provide ablation studies. We also show a robotic bin-picking experiment using descriptors learned from a fix-mounted camera for defining grasp preferences.

**Keywords:** self-supervised learning, computer vision, representation learning, bin-picking

## 1 Introduction

Scene and object understanding is essential for robot manipulation tasks, including assembly or bin picking. Often, the representation of choice is task-specific segmentation or pose estimation, trained in a supervised manner with labeled data. Labeling, however, is expensive and time-consuming, which is why self-supervised learning of dense visual descriptors has recently gained substantial attention in the robotics community, inspired by the works of [1] and [2].

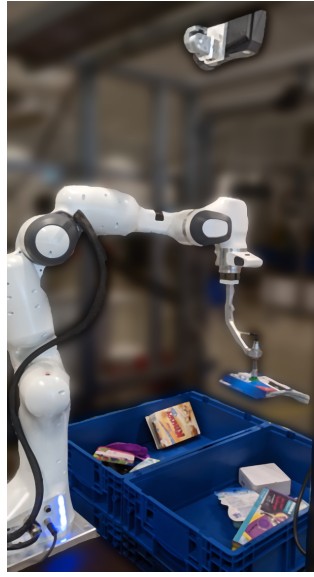

Dense Object Nets (DONs) learn dense visual descriptors of objects fully self-supervised in a robotic environment [2]. The learned descriptors are view-invariant, show potential for category level generalization and they naturally apply to non-rigid objects. The dense descriptor representation can be flexibly used for various downstream robotic tasks, such as, grasping [2, 3, 4], rope manipulation [5] and learning control [6, 7].

Self-supervised training of DONs, however, relies on pixel correspondences across multiple camera views provided by a registered RGBD image sequence, which requires accurate camera calibration and pose recording. Furthermore, pixel correspondence tends to be inaccurate with inexpensive depth cameras, even with data preprocessing. Finally, data collection is constrained by robot kinematics and the need for an expert setting up and supervising the procedure.

Figure 1: Bin-picking setup with a single, fix-mounted overhead camera.

6th Conference on Robot Learning (CoRL 2022), Auckland, New Zealand.

In this paper we relax these assumptions fundamentally and instead of a complex setup with a single, robot-mounted moving camera, or multiple static ones, we solely rely on an unordered set of RGB images to learn object descriptors, for example, recorded by a single fixed camera. In our work, instead of relying on multi-view, *geometric correspondence*, we use augmentations of single images to obtain alternative views and *synthetic correspondence*. This idea was already explored in computer vision [8, 9], in the context of learning geometrically consistent pixel-level descriptors across multiple object classes. In this paper we show that relying on synthetic image augmentations achieves competitive performance in terms of keypoint tracking accuracy compared to a network trained with geometric correspondence. Importantly, this approach can easily be adopted to existing industrial setups with fix-mounted cameras, or with cameras too heavy to be mounted on a robotic arm, without additional engineering effort. We show such a robotic bin-picking setup in Fig. 1, with an overhead, fix-mounted camera.

Our contributions are as follows: (i) we adapt existing work on training self-supervised pixel embeddings [9, 10] to robotic grasping downstream tasks. (ii) We show that for robotic grasping tasks our approach is en par with state-of-the-art [4] in terms of keypoint tracking accuracy while drastically simplifying the data collection. Finally, (iii) we show a real-world robotic bin-picking experiment where human preference on grasp configuration is encoded with dense visual descriptors, with the constraint of using a single, fixed-mounted camera.

## 2   Related work

In the following, we review recent work on self-supervised dense visual descriptor learning for robotic manipulation in more detail. We also discuss related work on self-supervised representation learning and learning from a set of single images, which are core concepts in our work.

**Dense visual descriptors in robotic manipulation.** Inspired by [1], Florence et al. [2] proposed self-supervised training of dense visual descriptors by and for robotic manipulation. Their approach was later adopted to learn from multi-view correspondence in dynamic scenes with an application for policy learning [7]. In [5] the descriptor space representation was applied to learn challenging rope manipulation in simulations. Applying dense descriptors for state representation in model predictive control learning was shown in [6]. In [11] multi-view consistency for keypoint learning was proposed with an application for state representation in reinforcement learning.

Another line of work investigated improved training strategies of dense visual descriptors. There are multiple contributions focusing on learning multi-object and multi-class descriptors [12, 13, 4]. The work in [3] exploits known object geometry to compute optimal descriptor embeddings. In [14] NeRF is utilized to generate dense correspondence datasets from RGB images. This alleviates problems with noisy depth data and proves especially helpful for thin and reflective objects.

There are several works that proposed to learn dense visual object representations directly from synthetic images composed of random backgrounds and randomly sampled, masked objects distributed over the image, see [2, 15, 12]. Learning from such synthetic images can be more efficient due to higher object density and ground truth correspondence, however, they rely on labeled datasets with object masks. Masking is either achieved by 3D reconstruction with a robot wrist-mounted camera [2, 15], or a labelled RGBD dataset [12]. As opposed to image composition via mask-labeled datasets, the image augmentation technique, as in this paper, only requires an unordered, unlabelled RGB dataset. This significantly simplifies data collection and opens up the possibilities to learn dense visual descriptors where no 3D reconstruction is possible, or where object masks are not available.

**Self-supervised descriptor learning from RGB images**. An intuitive way to generate geometric correspondence is to estimate the optical flow of subsequent frames from a video. In [16] this technique was adapted to train DONs using contrastive learning. [8] proposed to use optical flow from videos, or image augmentations to embed pixels of objects in view-invariant coordinate frames. This method was adopted in [9] for pretraining of geometry-oriented tasks, such as object specific part detection in images. The work in [17] learns pixel-wise descriptors from single images with augmentations by using hierarchical visual grouping of image patches based on contour. Our work follows the image augmentation technique of single RGB images to generate alternative views and synthetic correspondence. Equivariant network architectures (e.g., [18, 19]) could replace certain augmentations (e.g. rotation) during training and improve sample efficiency. In our setup we can

apply affine transformation on training data and we rely on a vanilla ResNet architecture for our experiments, which achieves good $SE(2)$ equivariance, as shown in our experiments.

**Self-supervised visual representations learning.** Instead of training on large supervised datasets, self-supervised methods have become a popular way to obtain visual representations, which can be fine-tuned to specific downstream tasks. A recent and very successful approach using contrastive learning is SimCLR [10]. It aims to maximize agreement between two augmented versions of the same image, while considering all other images in the batch as negative samples. BYOL (Bootstrap your own latent) [20], in comparison to contrastive methods, does not rely on the sampling of negatives. Barlow Twins [21] also forgo negative samples by optimizing the cross-correlation matrix between embeddings from two augmented versions of the same image to be close to identity. Our approach is most similar to SimCLR as we employ the same loss formulation, but with the important difference that our batch is constituted by individual pixel descriptors instead of full image embeddings.

## 3 Method

In this section we discuss our proposed training approach using image augmentations. We first give an overview of the whole training pipeline, then discuss image augmentation techniques and finally present the loss formulation and dataset requirements. For an illustration of the training pipeline we refer to Fig. 2.

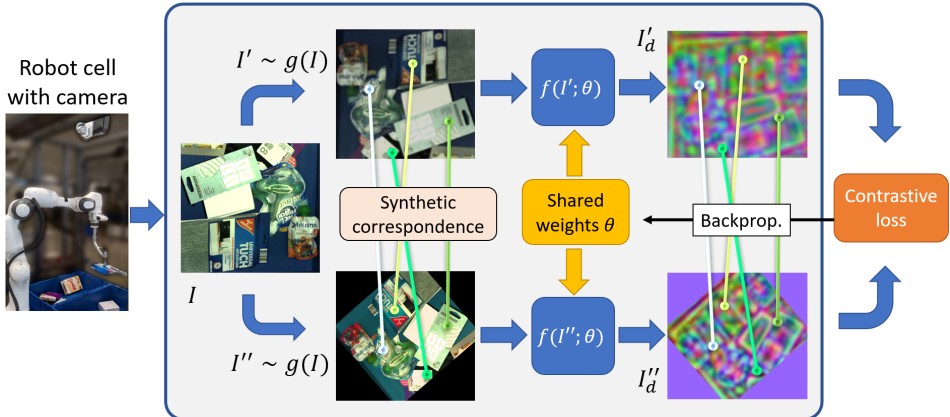

Figure 2: Illustration of the training pipeline. For every RGB image $I$ in a batch, we sample a pair of augmented images $I'$ and $I''$ while keeping track of the pixel correspondences. We evaluate the RGB images with the trained fully-convolutional network $f(\cdot; \theta)$ with trainable parameters $\theta$. Finally, we compute the contrastive loss and backpropagate the error using the embedded images $I'_d$ and $I''_d$ along with the correspondence information.

### 3.1 Dense Descriptor Training with Synthetic Correspondence

Inspired by the work of [9] we rely on training on an unordered set of images and use image augmentations to arrive at alternative views of each image. First, we sample a minibatch of $N$ RGB images from the training data set consisting of independent RGB images. For every image $I$ in the minibatch we sample two augmented views $I' \sim g(I)$ and $I'' \sim g(I)$ by applying randomized augmentations $g : \mathbb{R}^{H \times W \times 3} \mapsto \mathbb{R}^{H \times W \times 3}$ (described in more detail below). A learned fully-convolutional network [22] model $f(\cdot; \theta)$, $f : \mathbb{R}^{H \times W \times 3} \mapsto \mathbb{R}^{H \times W \times D}$ maps the augmented images $I'$ and $I''$ to their descriptor space embeddings $I'_d$ and $I''_d$. The user defined parameter $D \in \mathbb{N}^+$ controls the resolution of the descriptor space.

By keeping track of the position of each pixel in the original image $I$ during the augmentations, we sample pairs of pixel locations between $I'$ and $I''$ that share the same position in $I$. We refer to these as *synthetic* correspondences, emphasizing the use of synthetic image augmentations as opposed to *geometric* correspondences coming from the 3D geometry of multiple camera views, as

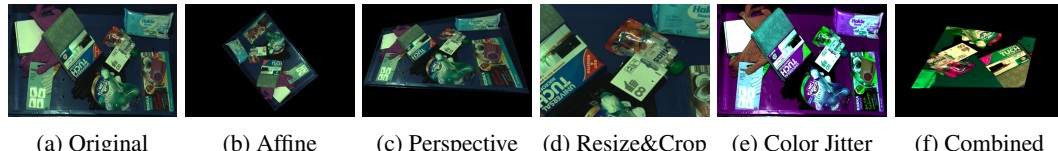

| (a) Original | (b) Affine | (c) Perspective | (d) Resize&Crop | (e) Color Jitter | (f) Combined |

Figure 3: Visualization of the augmentations utilized for synthetic view training. Figure 3f shows the combination of all augmentations as used in practice for training a synthetic view descriptor model. While each augmentation is guaranteed to be applied, the individual parameters, e.g., scale of distortion, angle of rotation, crop size and location, etc. are still randomly selected each iteration.

in [1, 2]. The descriptor values at the sampled pixel correspondence locations serve as positive pairs for contrastive learning.

## 3.2 Image Augmentations

For robotic applications we require descriptors that are invariant to translations, rotations and perspective changes of the objects, as well as changes of lighting conditions. In vanilla DONs training, cf. [2], this is achieved by recording a diverse training set of registered RGBD image sequences, which contain sufficient variance in camera and object poses, and to some extent in lighting conditions. In our work, we achieve a similar effect purely by imposing data augmentations on single RGB images from an unordered set.

We carefully select augmentations, which reflect the desired invariance properties stated above, as follows: *affine transformation* induce rotations and scale changes (zoom out), *perspective distortions* mimic view changes, *resize&crop* implies scale changes (zoom in), and lastly, *color jitter* affects brightness, contrast, hue, and saturation of the image, hence the lighting conditions of the scene. We show an illustration of these augmentations in Fig. 3. We utilize *torchvision library* [23] of Pytorch as reference implementations.

The positive impact of image augmentations on the training of dense visual descriptors was already demonstrated in [4] for datasets utilizing geometric correspondences. In our work, each augmentation is not only helpful, but relevant to the ability of the model to successfully learn an invariant descriptor space. See Appendix F.1 for an ablation on the respective impact of each augmentation on the overall performance.

## 3.3 Loss Function

Following [10] we adopt the NT-Xent loss, for learning the dense descriptor representations. For a pair of corresponding descriptors $\{d_i, d_j\}$, obtained from images $I'$ and $I''$ in a minibatch of size $M$, we compare their distance to the distance of $d_i$ to all other sampled descriptors in the given minibatch arriving at the following individual loss term:

$$l_{i,j} = -\log \frac{\exp(\text{dist}\left(d_i, d_j\right)/\tau)}{\sum_{k=1;k\neq i}^{2M} \exp(\text{dist}\left(d_i, d_k\right)/\tau)}. \tag{1}$$

with temperature parameter $\tau$ which we fix to $\tau = 0.07$ throughout this paper following [4]. We choose the metric dist $(\cdot)$ to be the cosine similarity between descriptors $d_i$ and $d_j$. Due to the cosine similarity we normalize the descriptors $d$. The total loss is given as the mean over all individual loss terms, cf. [10].

Note that this loss, together with our correspondence sampling, does not distinguish background from objects, nor does it explicitly address multiple object classes and instances, as opposed to [2]. Instead, we follow [4] and sample correspondences uniformly in image plane assuming that every pixel is unique. This method is tailored to datasets depicting densely packed scenes with single object instances, for example, a heap of objects in a bin-picking scenario. The learned descriptor space does not imply semantic information on object classes or background, but still provides consistent keypoint detection and robust tracking performance, which is essential for downstream tasks.

# 4 Comparison of Training with Synthetic and Geometric Correspondence

In this section we show an in-depth comparison between training with geometric and synthetic correspondence. We also investigate the invariance of descriptors obtained from a network trained with synthetic correspondence with respect to object-camera relative transformations.

Similar to [2] we use a pretrained ResNet-34 with 8-stride output for all experiments. The upsampled output matches the resolution of the input. In Appendix B we give more details on the baseline training method using geometric correspondence.

We recorded a dataset of registered RGBD sequences. Despite the availability for registered image pairs, the synthetic training only uses single RGB images for both training and validation. However, the camera poses help with the evaluation as they allow us to generate ground truth pixel matches across any two images of the same static scene without the need for manual labeling. The dataset consists of eight scenes with various object configurations. Every scene contains only one instance per object. The scenes are recorded with a robot wrist-mounted camera while the robot arm follows a predefined trajectory keeping the camera view on the objects.

Both approaches are evaluated on the same ground-truth image pairs and correspondences. For robustness, we perform a k-fold cross-validation, that is, each scene from our total dataset was once used as test set. One scene is chosen as validation set, with the remaining six scenes used for training. We report the averaged results. We use the same loss function, training parameters and augmentations for both approaches, with the exception that augmentations are chosen with $50\%$ probability for the geometric training, as it yields better results. For synthetic training, each augmentation is applied. An ablation study of using different augmentation probabilities is given in Appendix F.2. For brevity, we report the median and 75% quantile pixel errors. Extended results, including the mean, 90%, and 95% quantile pixel errors, as well as the percentage-of-correct keypoints@k (PCK@k) for $k \in \{3, 5, 10, 25, 50\}$, are reported in the respective Appendix sections.

## 4.1 Keypoint Tracking Performance

Given a common dataset, measuring keypoint trackingThen, wery descriptor $d_A$ in image $B$ and record the pixel error $e = \|k_i^B - k_i^*\|_2$. The network parameters $\theta$ are either obtained by geometric [4] or synthetic correspondence training proposed in Sec. 3.1.

**Results.** Fig. 4a shows the distribution of pixel errors for the two training approaches. The median of the corresponding distribution is highlighted as a dashed line. With a difference of only 1.9px in median pixel error, the synthetic correspondence performs competitively to the geometric training. The percentage of pixel errors that are larger than 50 pixels are $10.1\%$ and $5.4\%$ respectively, as indicated in the bottom right part of Fig. 4a.

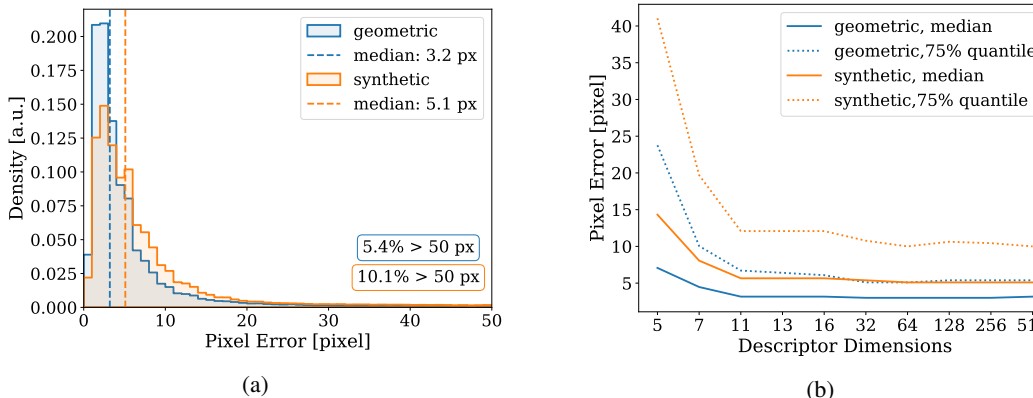

(a)                                                                (b)

Figure 4: (a) Pixel error distributions obtained using synthetic (orange) and geometric (blue) correspondence training. (b) Median and 75% quantile of the pixel error distribution for both training approaches and different descriptor dimensions.

In Fig. 4b we compare the median and the 75% quantile of the two training methods with respect to the descriptor dimensions $D$. These results were obtained without k-fold cross-validation. For both training approaches the median pixel error decreases for larger descriptor dimensions. For dimensions larger than 9 the median pixel error decreases only marginally. In contrast, the 75% quantile error still increases further until $D = 64$, before improvements saturate. For additional insights into these results we refer to the supplementary material.

## 4.2 Invariance Tests

In the following, we discuss the question: how well does training with image augmentations proposed in Sec. 3.2 generalize to physical camera transformations? We recorded different test scenes with a wrist-mounted camera and the following specific camera movements: (i) changing camera perspective (camera tilting), (ii) translation along the camera z-axis (zooming in and out), and (iii) camera rotation along the camera z-axis (see Appendix D.2 for details). We compare the performance of a network trained with synthetic and one with geometric correspondence both trained on the same data as described in Sec. 4.1. We use 64 descriptor dimensions and affine, perspective and resize&crop augmentations. As in the previous section we compute the 75% quantile pixel error and use 1000 keypoints per image pair, fix the base image $A$ and only vary image $B$, which shows the changing camera views. The results for three different types of transformations are compiled in Fig. 5.

It can be seen that training with synthetic correspondence generalizes well to a large range of camera transformations, especially to those parallel to the object plane (Fig. 5a, Fig. 5b). These physical camera transformations are very similar to the affine and resize&crop augmentations used during training. For generalizing to the perspective transformations with angles above $45°$ degrees, as shown in Fig. 5c, the synthetic correspondence training shows a clear deterioration in performance. As the perspective changes, occluded parts become visible and vice versa. This physical transformation is not well captured by the synthetic augmentations for larger angles.

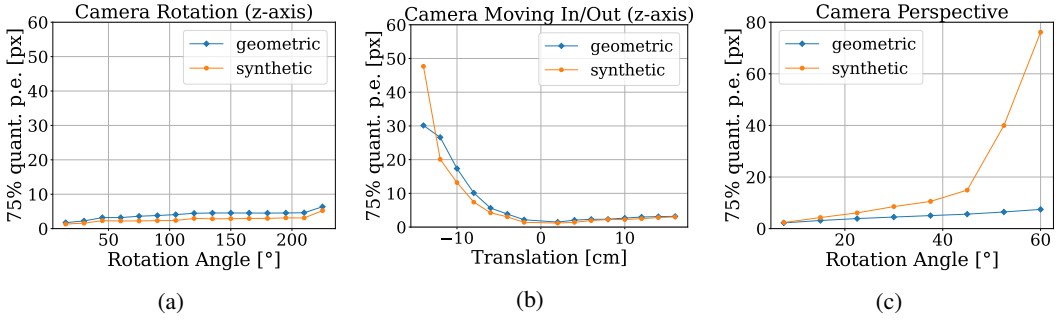

(a)  (b)  (c)

Figure 5: 75% quantile pixel error between two different images for the geometric (blue) and synthetic (orange) correspondence training approach for three different isolated camera movements between those images in varying magnitude: (a) camera rotates around z-axis, (b) camera moving closer and further away from the objects, (c) camera moving on a sphere in x-direction around the objects, facing the objects. Note the scale of the y-axes.

## 5 Grasping Experiment with Fix-mounted Camera

We demonstrate a robotic bin-picking experiment that relies on dense visual descriptors for defining grasp preferences. We use a 7-DoF Franka Emika Panda arm with a suction gripper mounted on the end-effector, see Fig. 1. Our setup uses a fix-mounted Zivid One+ camera above the bin in a robotic cell. Training a descriptor network using this setup prevents the use of geometric correspondences. Instead, we show that our proposed method can be trained on a setup as it is often present in real world applications and prove that the keypoint tracking accuracy is good enough for guiding a generic grasping method by human annotated grasp preferences.

In the experiment, we consider picking ten different types of objects from one bin. However, using a suction gripper the objects are difficult to grasp at certain locations: some have cutouts on the

packaging, transparent or foldable parts, and uneven surfaces (see Appendix G for higher resolution images). In our experiments, a purely model-free grasp pose generator often predicts poses at these locations, ultimately reducing picking performance.

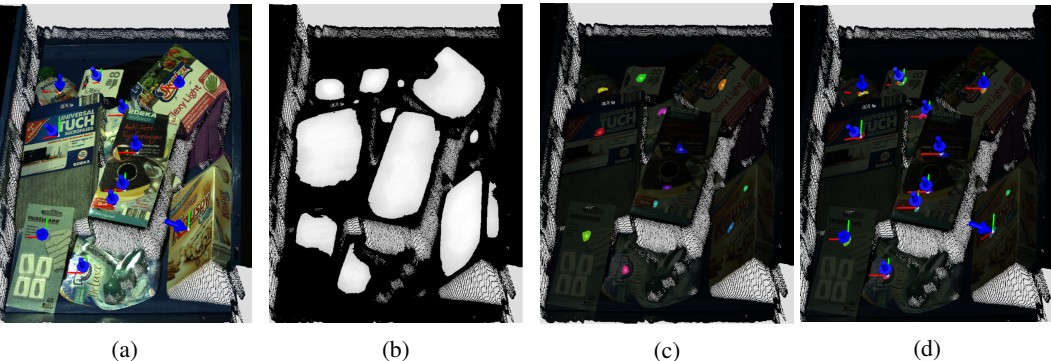

(a)  (b)  (c)  (d)

Figure 6: Illustration of the grasping experiment. (a) The RGB images with the predicted grasp poses. (b) The graspable areas generated using RGB and depth images. (c) The predicted keypoint heatmaps on graspable areas. (d) The predicted grasp poses overlayed on the heatmaps.

We want to utilize human domain knowledge to aid grasp pose selection using dense visual descriptors to increase grasp success rates for these objects. To do so, we show a small number of RGB images depicting the objects in different configurations in the bin and ask the human to click on pixel locations corresponding to a preferred grasp location. We track these descriptors to generate a *preference heatmap*, as shown in Fig. 6c. The heatmap, in contrast to discrete single-pixel correspondences, enables intuitive consideration of matching uncertainty in the form of distance in descriptor space. Furthermore, it provides a more flexible and quantifiable basis for combination with grasp detectors.

To generate the final set of grasp pose candidates we intersect the grasp preference heatmap with the detected graspable areas identified by the model-free grasp detector (Fig. 6b). While we make no specific assumption regarding which grasp detection method to use, we employ a dense pixel-wise graspability estimation in this experiment. It is based on a fully convolutional neural network with RGB-D input, specifically UNet [24] trained on annotated pixel-wise labels of expected graspability for a wide range of different bin picking scenes. For the example image, the resulting poses are shown in Fig. 6a and Fig. 6d.

Note that the descriptor network is trained purely from the set of RGB images showing the bin with objects in random configurations, recorded with the overhead camera. See Sec. G for more details on the experimental setup and heatmap generation.

## 5.1 Quantitative Evaluation

We evaluate the benefit of the proposed method for robotic bin picking based on a set of sixteen manually annotated scenes. The scene images include the ten known objects, but in different configurations compared to the training set. For each evaluation image, we manually annotated object instances, pixel-wise graspable areas, and those areas that correspond to selected descriptors. In total, the evaluation dataset contains 131 graspable objects of which 103 have visible descriptors.

We compare our descriptor-based grasping approach with a purely model-free approach that directly uses the graspable areas without accounting for descriptors. With this study, we investigate two questions. First, how effective is our approach to re-identify descriptor spots compared to grasping at these spots by chance? And second, is there a considerable negative effect on the amount of objects that can be grasped, e.g., due to non-detected grasp pose annotations?

Table 1 shows a summary of the results. *Success Rate* denotes the number of successful grasps compared to all grasps attempted. *Descriptor Success* denotes the number of grasps at those spots marked as desired grasping points compared to all grasps attempted. *Object Hits* denotes the percentage of all graspable objects for which at least one feasible grasp pose has been found irrespective of descriptors, including objects without visible descriptor spots. *Descriptor Hits* denotes the per-

centage of objects for which a grasping pose has been found at the respective descriptor spot, only including objects with visible descriptors. We consider descriptor spots with a tolerance of around 1cm, which corresponds to the radius of the suction gripper.

Table 1: Evaluation results on individual, annotated scene images similar to Fig. 6.

|  | Success Rate | Descriptor Success | Object Hits | Descriptor Hits |
|---|---|---|---|---|
| Preference (ours) | 98.9% | 91.1% | 63.4% | 78.6% |
| Baseline | 79.9% | 50.4% | 78.6% | 68.0% |

It can be concluded from Table 1 that using the proposed method to encode grasp preferences is effective as it significantly increases the amount of grasps at desired spots from $50.4\%$ to $91.1\%$. Due to the challenging object geometries, this helps to raise the overall grasp success rate from $79.9\%$ to $98.9\%$. The expected downside, however, is that the proposed method finds grasp poses only for a smaller amount of objects, $63.4\%$ instead of $78.6\%$. Still, this includes objects that have no desired grasping spot visible. In some applications it can be the desired behavior to not propose a grasp pose for objects if they cannot be grasped at the preferred location. When only considering which preferred grasping spots have been covered by grasp poses, our method manages to find grasps for $78.6\%$ instead of $68.0\%$ of the visible spots. In consequence, we conclude that there is only a moderate negative effect due to limiting grasping to descriptor spots which can indeed be beneficial for some applications.

## 6 Limitations

**Variance in camera poses.** As seen in Fig. 5, the synthetic training ensures that descriptors are stable within a limited margin of camera transformations relative to the object. For example, in Fig. 5c, for viewing angles steeper than $45°$, accuracy deteriorates. This imposes a limit on descriptor consistency across images with large differences in object poses.

**Changing Environment.** A change in environment (background, lighting) between training and inference time may have a negative influence on keypoint tracking performance. It is expected that augmentations such as color jitter and, if masks are available, background randomization can mitigate these effects. We will investigate these aspects in future research.

**Generalization to unseen objects.** Although [2] demonstrated capabilities to generalize intra-class instances, our work focuses on instance specific descriptors. Given our outlined training setup, it works best on known objects from the training set.

**Object Edges.** We observe worse descriptor consistency across images for keypoints located at the edges of objects or close to parts that are occluded by other objects. In setups where object masks are present background randomization could reduce this effect.

## 7 Conclusion

In this paper we proposed a novel training method for learning dense visual descriptors based on image augmentations for robotic manipulation. The evaluation shows that overall our proposed method is competitive to the existing geometric training approach. For physical transformations like changing the camera perspective on the scene, which are harder to mimic by augmentations, the training method using geometric correspondences shows superior performance. Being aware of these limitations our proposed method is expected to perform well for setups where objects are mostly altered by translations, rotations parallel to the camera plane, or are slightly tilted. As this is often the case for random heaps of objects in a bin, our method is especially suitable for such setups that are constrained by a fix-mounted camera. Finally, we demonstrated the use of our method in a realistic grasping experiment to increase grasp success rates by human annotated grasp preferences.

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
