# OpenReview forum: "Learning Dense Visual Descriptors using Image Augmentations for Robot Manipulation Tasks"
_robot-learning.org/CoRL/2022/Conference — CoRL 2022 Poster_

### Official Review · Reviewer_aGSf · 2022-07-17

**Originality:** Fair
**Technical Quality:** Good
**Clarity Of Presentation:** Good
**Impact:** 3

**Recommendation:**

Weak Accept: I recommend accepting the paper, but will not argue for my recommendation if the majority of other reviewers have a different opinion.

**Summary:**

The authors propose  a way for dense correspondence descriptor learning that can learn these descriptors from an unordered set of RGB images which can come from a single stationary camera.  They achieve this by transforming the input image with various data augmentations, keeping track of the ground truth correspondences. They then study the performance of this training regime compared to more standard dense descriptor learning that requires self-supervised data gathering with  registered cameras from different viewpoints.

**Issues:**


- Perhaps the authors can better justify why the specific approach/design choices of synthetic training they used in that paper are interesting in their own right, compared to the other approaches using synthetic training for dense correspondences (for example in Learning Multi-object dense descriptor for autonomous goal-conditioned grasping by Yang et al.)?
- Similarly perhaps the authors could expand on what additional insights do the particular robotics experiments conducted in this paper bring, given that previous works have already demonstrated multiple times that dense descriptors are well suited for finding picking locations?
- line 79 - Title of this section seems a bit odd  with the phrase that follows: If you use a video and optical flow between subsequent images, then it is not descriptors from ``single images’’?
- line 132 - It is unclear what ``geometric datasets’’ refers to?
- Fig. 5 : median is more intuitive, if 75% quantile is needed instead, then it would be good to mention why?
- In section 4.2 it is unclear whether the same datasets where used as in section  4.1?
- In related work it would be nice to situate more explicitly this work with the previous works in the field, where does it draw inspiration and how, and where is it different. This information is currently scattered across the paper, it would be nice it if was made clearer in the related work.
- The loss in eq. (1) is quite central,  could be nicer to have a more complete explanation of it rather that simply refer to previous work : What is M and tau in the loss? What is the intuition behind it?
- Line 154 is a bit unclear:  The phrase as written reads as if the cosine similarity loss was doing the normalisation of the vectors. I believe this is not what was the intended meaning, descriptors can be normalised because the cosine similarity doesn’t impose anything  on the norm of the descriptors, but the normalisation is done independently from the cosine similarity?
- line 170 : It is not clear what the augmentations are: are those the same augmentations as with the synthetic correspondences?
- Are the training and test scenes constrained to have the same objects present? This should be made clearer.

**Quality Of The Limitations Section:**

Limitations are addressed clearly

**Reviewer Expertise:**

4: The reviewer is confident but not absolutely certain that the evaluation is correct

**Robotics Focus:**

Sufficient demonstration on hardware

**Strengths And Weaknesses:**

**Summary of Strengths**
- Fair Literature review.
- Approach is clear and well presented.
- Good experimental procedures and analysis.

**Summary of Weaknesses**
- Novelty in the approach somewhat limited.
- Analysis suffers from having a single baseline.
- It is unclear why the grasping experiment is necessary.

**Comments**
- The idea of using synthetic data to train dense descriptors does not seem new (for example Learning Multi-object dense descriptor for autonomous goal-conditioned grasping by Yang et al.), nor does the specific loss used  (coming from Chen et al. A simple framework for contrastive learning of visual representations ). Moreover, the synthetic training from Yang et al. seems like it would provide the same benefits over vanilla dense object nets as the proposed approach, and the proposed approach generally does not perform better than the geometric baseline.It seems to be this work would benefit from expanding on the analysis side. That is, benchmarking more approaches  that are interesting and appear in different previous works, with synthetic or geometric training, and analyse the pros and cons of each approach to build a more wholistic study. This would then result in a very interesting analysis paper, as opposed to proposing a specific approach.

- Using dense descriptor representations for finding human-defined grasp locations in different scene configurations has been shown to work well in the papers from the related work. Showing that using dense descriptors to guide a model-free grasping method seems to have an incremental benefit over this, which distracts from what otherwise seems to be the main point of the paper which is a comparison between the ``standard’’, geometric , way of training dense correspondence descriptors and the synthetic way of training them.



**Summary Of Recommendation:**

The paper has limited novelty in the method but seems well executed and presented. I believe a much stronger paper could arise from it by further developing the analysis side,  by comparing over multiple baselines and approaches, rather than favouring any particular method as the "proposed approach". That is casting the narrative as "providing insights and finding the pros and cons of different approaches for learning dense correspondence descriptors", and comparing various interesting approaches (including those already analysed) to provide those insights.

---

> ### Author Response · Authors · 2022-08-24
> **Answers to Reviewer 4 (aGSf)**
>
> We thank the reviewer for the useful comments and suggestions. Please see our comments below.
>
> **[4.1] Novelty of approach:** Previous approaches in dense descriptor learning for robotics (Florence et al.[2], Yang et al.[12]) generate synthetic images relying on object masks and class labels. Class labels need to be annotated and object masks require at least a 3D reconstruction including scene registration, or an annotated dataset.
>
> As opposed to these approaches, we rely only on a set of RGB images collected by a single RGB camera, without depth image, registration, object masks or class labels. This prevents us from composing synthetic images and pixel correspondences based on class labels and masks as in previous approaches, e.g., by Yang et al. [12].
>
> In our approach we augment RGB images to generate alternative views and pixel correspondence. Without object labels and masks, we obtain object descriptors implicitly by learning the whole scene represented in the RGB images. We would like to emphasize that this allows for a significant simplification of the data collection pipeline (only need RGB images), that is well suited to learning object descriptors of specific, task relevant objects.
>
> We updated the Related Work section to outline this difference more clearly, see rebuttal pdf section V for the changes made.
>
> **[4.2] Grasp-preference experiment:** We agree that it is state of the art to use dense descriptors for finding human-defined grasp locations and improved the experiment description by highlighting the challenge compared to other settings, i.e., using geometric correspondences for training is not possible in our setup.
>
> Consequently, we see it as an important experiment to show that we can train and apply our proposed method for human defined grasp preferences under the strong assumptions of a standard robotic cell without the need for additional camera or other hardware to collect training data.
>
> Further it is important to show that the results we obtained from our keypoint tracking experiments (Sec. 4) do translate to a more challenging environment. While in Sec. 4 we measured keypoint tracking accuracy among different views in the same scene, in the robot grasping experiment we show that our method generalizes to detecting keypoints in different object configurations including occlusion.
> We improved the description of the grasping experiment and highlighted our motivation more clearly.
>
> **[4.3] Metrics:** We focused on the 75% quantile primarily to simplify figures and because the median does not reflect the tail of the pixel error distribution. Furthermore, it highlights the trend of the distribution on the invariance tests well. We added this explanation to the text and added a table including more metrics such as mean, median, 75%, 90%, 95% and several PCK measures to the appendix for further insights, see rebuttal pdf Section III for results.
>
> **[4.4] Object generalization in grasping experiment:** The statement about the objects in the test set was somewhat misleading. The “not contained in the training set” referred to the specific images, not the objects itself. We used the same objects, but in different configurations in the scene. We improved the description in the paper.
>
> We added another experiment to explicitly test the generalizability, see details in comment 2.1 (Wx5g). In short, for “easy” examples the methods generalize well to other object, but for strong perspective changes we need in-class training data; cf. rebuttal pdf Section III.2 for results.
>
> **Minor Comments:**
> - Updated title of subsection in Sec. 2 (previously line 79)
> - Rewrote sentence with ‘geometric datasets‘ (previously line 132)
> - Specified the used datasets in Sec. 4.2
> - Expanded discussion of loss function in Sec 3.3
> - Clarified sentence about cosine similarity (previously line 154)
> - Specified that we use the same augmentations as with the synthetic correspondences

---

> > ### Comment · Reviewer_aGSf · 2022-08-26
> > **Good Clarifications - will be upgrading the recommendation.**
> >
> > I thank the authors for their clarifications. My main concerns were addressed appropriately, so I will be upgrading my recommendation accordingly.

---

### Official Review · Reviewer_etoZ · 2022-07-19

**Originality:** Fair
**Technical Quality:** Good
**Clarity Of Presentation:** Good
**Impact:** 3

**Recommendation:**

Weak Accept: I recommend accepting the paper, but will not argue for my recommendation if the majority of other reviewers have a different opinion.

**Summary:**

This paper proposes a data augmentation strategy of RGB images for self-supervised visual descriptor learning. Previous approaches require geometric correspondences derived from image registration via e.g., reconstructions of video sequences, to compute a contrastive (or variant of) loss during training. The presented approach simply augments a single RGB input image and learns an expressive pixel feature representation through the proposed "synthetic correspondences", i.e., same pixel after undergoing image augmentation. Experiments show that learning from synthetic correspondences achieves similar performance to learning from geometric correspondences in terms of keypoint matching. The method is used to demonstrate a robotic grasping task in which hand labelled grasp locations on objects can be identified in new scenes through pixel descriptor matching.

**Issues:**

As discussed in "weaknesses", please clarify the main technical contribution in light of existing work on the same topic as well as provide much more detail about the robot grasping experiments. Other minor issues that should be addressed are as follows:

* In Sec. 4 (lines 170-171), it is explained that for geometric correspondence training, the augmentations are chosen with 50% (instead of 100% for synthetic correspondence training). There is no explanation, conjecture or quantitative results to support the statement in the text or supplementary. It would be good to add an analysis not only of this aspect but that 100% augmentation is the best for learning from synthetic correspondences. Interestingly, [14] report that augmenting only one of the images leads to better performance, so it seems that the "right" amount of augmentation is inconclusive.

* Sec. 4.2. could be made more clear with a diagram to explain the camera movements as it is not immediately clear from the text - we have to make a strong assumption about the camera coordinate system. The diagram can go in the supplementary and complement Fig. 8.

* The work does not explicitly address generalisation as the dataset considers the same objects in both testing and training. I was surprised that the robot manipulation experiments include unseen objects in the evaluation and that the descriptor matching appears good (in terms of descriptor success). Could the authors comment on why this is the case? Are the test objects similar to those seen during training?

**Quality Of The Limitations Section:**

Limitations are addressed clearly

**Reviewer Expertise:**

4: The reviewer is confident but not absolutely certain that the evaluation is correct

**Robotics Focus:**

Sufficient demonstration on hardware

**Strengths And Weaknesses:**

## Strengths

The problem of learning visual descriptors is important and highly relevant for many applications such as robot manipulation as demonstrated by the authors. The authors have also covered the relevant literature sufficiently.

The technical approach is sound with a mostly good description of the approach (although there are some details missing as discussed later).

The experiments demonstrate that the proposed approach achieves good success in comparison to alternatives. They especially highlight the capability of good keypoint tracking without the requirement of generating geometric correspondences. Lastly, the grasping experiments make this paper highly relevant to the themes of CoRL.

## Weaknesses

The main weakness is that the authors have not strongly asserted their technical innovation in light of the SotA. The use of augmentation has already been shown in various papers, e.g., [4, 9, 10]; furthermore, the importance of the different augmentations, namely the affine transformation, have been demonstrated to achieve more robustness. It is argued that other work, e.g. [14], learn image embeddings instead of pixel descriptors but I do not see this as a major technical contribution as it translates to an adjustment of the loss, which is ultimately borrowed from [10], as well as task and metric for evaluation.

I find the grasping experiments poorly explained. What is the model-free grasp pose generation method? There is no explanation (not even in the supplementary) and I can only glean that it detects uniformly flat surfaces from Fig. 7b. Also, why does the final grasp need to be the intersection between the graspable areas and the grasp preference heat map? Who couldn't the raw user-selected pixels from one labelled image be used without having to first compute a heatmap? The original work in [2] shows this capability precisely. The descriptor success and hits metrics are not entirely fair for the baseline since it is completely agnostic to the descriptors. I am actually rather curious why the baseline achieves decent scores but it is hard to reflect on this without knowing more about the underlying method.

I have a minor issue with the metrics used in the experiments due to a lack of clarity. Many of the results are only expressed in terms of the 75% quantile without justification of why this is the best metric. Why was the median dropped after results in Fig. 4? Furthermore, most related work report PCK, as such, I think the authors should argue why their choice of metric is more appropriate than what appears to be the standard.

**Summary Of Recommendation:**

While the paper presents an interesting and simple idea to learn visual descriptors, as well as demonstrate this for a robotic application, the technical contribution is relatively low as similar ideas have already been presented in previous work. The main difference appears to be the application of a pixel-wise contrastive loss from [10] for the setting proposed in [14] in order to learn pixel descriptors rather than image embeddings. This is only incremental in terms of contribution. Furthermore, it is difficult to assess the significance of the robotic grasping experiments without more information regarding the model-free grasping method (the performance comparison is unclear without a better understanding of the baseline). Lastly, the metrics used for evaluation are unjustified and differ from prior work. As such, I recommend a weak reject but am open to see a revision that potentially remedies these flaws.

---

> ### Author Response · Authors · 2022-08-24
> **Answers to Reviewer 3 (etoZ) [Part 1/2]**
>
> We thank the reviewer for the useful comments and suggestions. Please see our comments below.
>
> **[3.1] Comparing against SoTA:** We agree with the reviewer that in the original submission we didn’t highlight the differences to the SotA clearly. As the reviewer points out, our method is a straightforward combination of the single image augmentation method for descriptor learning by Novotny et al. [9], the augmentation techniques for the robotics domain by Adrian et al. [4] and the loss by Chen et al. [10]. The techniques presented in our paper are not novel compared to that of [4, 9, 10] and others.
>
> However, we believe the combination of the above methods for the robotics domain, especially in existing industrial robot cells with a fixed-mounted camera, where generating geometric correspondence is not possible, is novel. To the best of our knowledge our paper is the first one that thoroughly evaluates this approach and compares it to the geometric correspondence alternative presented by Adrian et al. [4], Florence et al. [2], or Yang et al. [12].
>
> The main message of our paper is that with the proper augmentations we can achieve competitive performance to that of working with geometric correspondence, but with a drastically simplified data collection pipeline (capturing RGB images), which can directly generalize dense visual descriptor learning to existing robotic environments without any engineering effort.
>
> We have expanded the introduction and related works section and clarified our contributions, see the rebuttal pdf Section V.
>
> **[3.2] Explanation of grasping experiment:** The initial explanations of the model-free gasping method was indeed lacking details and we now extended the explanations. Generally, we make no assumptions on which method to use, as long as it can incorporate the generated heatmaps. For this reason, we found it most intuitive to rely on a dense pixel-wise grasp detection and use a fully convolutional neural network to predict it.
>
> The purpose of computing a heatmap is mainly for robustness reasons. While it is possible to directly grasp at the pixel with smallest descriptor distance to the selected one, the heatmap based on descriptor similarity gives more flexibility to account for graspability predictions, essentially finding the best compromise between a grasp being both feasible and desired. In addition, it better reflects matching uncertainty than just considering the discrete maximum.
>
> We agree that comparing the descriptor success and hit metrics with the baseline method is not entirely fair. However, it is also not to be understood as a comparison in the sense which method performs better, but rather a sanity check for our proposed method to see how high these values would already be without a contribution of the descriptor matching. This has been clarified in the updated version of the paper.
>
> The fact that, e.g., around 50% of the grasps proposed by the model-free method are already close to a selected descriptor is most likely due to the descriptors being selected at preferred grasping locations, sometimes similar to what a model-free method can find as well implicitly. Thus, it is an important comparison in our view to see how many of the around 90% grasp poses being close to descriptors as found by our method are actually coming from descriptor matching and would have missed by solely detecting feasible grasps.
>
> **[3.3] Metrics:** Thank you for highlighting this shortcoming in our evaluation. We focused on the 75% quantile primarily to simplify figures and because the median does not reflect the tail of the pixel error distribution. Furthermore, it highlights the trend of the distribution on the invariance tests well. We added this explanation to the text.
>
> In other work, however, it is indeed common to either report the (normalized) mean pixel error, or some selected PCK@k metrics. We now also provide a full table of metrics in the appendix, cf. rebuttal pdf Section III. The complete set of metrics  now include the mean and median pixel error, 75%, 90% and 95% quantiles, as well as the PCK@k metric evaluated for k=[3,5,10,25,50]. Unlike other work, we chose not to evaluate PCK@k for higher k-thresholds, as we consider offsets of more than 50 pixels as irrelevant for any robotic manipulation application.

---

> > ### Author Response · Authors · 2022-08-24
> > **Answers to Reviewer 3 (etoZ) [Part 2/2]**
> >
> > **[3.4] Probability of augmentations:** We agree that the choice appears without explanation. We now provide the underlying additional results for this choice as new ablation study in the paper, with experiments conducted for both the geometric and synthetic correspondence training. See rebuttal pdf Section IV.1.
> >
> > Both approaches work well with augmenting either one or both views. However, the amount of augmentation applied is more critical. In particular, our training needs to have at least one image augmented, hence guaranteeing augmentation of both views works well in practice. For geometric correspondence training we reconfirm the results of Adrian et al [4], that 50% works best and, e.g., augmenting both views with 100% degrades performance. Most likely, choosing augmentations like color jitter versus affine transformations with a different chance, could further improve results.
> >
> > **[3.5] Explanation of camera movements:** We agree that this needs further explanations. For this we added a section to the appendix, the content can be found in rebuttal pdf Section II, explaining the dataset used for the invariance tests from Sec. 4.2 in more detail and further added a figure visualizing the coordinate system and the specific camera movements to the appendix.
> >
> > **[3.6] Generalization:** A study on generalization to different objects was indeed missing and we now added such an evaluation to the paper. As this concern was also raised by other reviewers, find more details above in the answers to reviewer Wx5g, comment 2.1; cf. rebuttal pdf Section III.2 for results.
> >
> > **[3.7] Objects in grasping experiment:** This statement was somewhat misleading. The “not contained in the training set” referred to the specific images, not the objects itself. We used the same objects, but in different configurations in the scene. We improved the description in the paper.

---

> > > ### Comment · Reviewer_etoZ · 2022-08-26
> > > **Response to authors' answers [2/2]**
> > >
> > > __[3.4]__, __[3.5]__, __[3.7]__ All clarified.
> > >
> > > __[3.6]__ The results are interesting and reveal that generalisation is problematic as all metrics show a notable decrease in performance (I disagree with the statement that "good average performance indicated by the median or the PCK metrics").

---

> > > > ### Author Response · Authors · 2022-08-26
> > > > **Answers to Reviewer 3 (etoZ) [Part 2/2]**
> > > >
> > > > **[3.6]** We agree that the referenced paragraph is unclear about the implications. We updated the section as below:
> > > >
> > > > > Both approaches exhibit a loss in performance, especially the geometric correspondence training.
> > > > > While the median changes only slightly, we find a large increase with respect to the 90% and 95% quantile for both methods.
> > > > > Up to 25% of the sampled keypoints are now mispredicted with an error nearly three times as high as before.

---

> > ### Comment · Reviewer_etoZ · 2022-08-26
> > **Response to authors' answers [1/2]**
> >
> > __[3.1]__ Thank you for the deeper insight: it is more clear that your work presents a novel combination of methods with a focus on downstream robotic tasks. The modifications in the attached .pdf make a stronger case for the contribution and highlight the practical utility of the technical work. This will be taken into consideration when evaluating the originality criterion, i.e., the value of a new method vs. the value of a new combination of existing methods.
> >
> > __[3.2]__ The overview is good but this still lacks detail. What is the architecture exactly (e.g., number/size of layers, etc.)? What is the ground truth data used to train on? Is this method based on any previous work?
> >
> > It would be interesting to see some results for grasping directly on the smallest descriptor distance. But it is true that a heatmap helps to deal with the matching uncertainty and I agree this is more robust.
> >
> > __[3.3]__ Good explanation and the results for the metrics are a welcomed addition. I still suggest to report mean pixel error and PCK prominently in order to be consistent with prior work.

---

> > > ### Author Response · Authors · 2022-08-26
> > > **Answers to Reviewer 3 (etoZ) [Part 1/2]**
> > >
> > > **[3.2]** We further refined the description of the grasping method. To summarize: We used an off-the-shelf UNet in the experiment and now added a reference to the paper for increased clarity. Other than that, the method is not based on specific previous work.
> > >
> > > Nevertheless, there exist indeed various approaches for predicting pixel-wise graspability in the literature which we think could work well with the presented heatmap approach, but this is beyond the scope of experiments we did for this paper.
> > >
> > > Regarding ground truth data, the supervised training has been given annotated pixel-wise labels for expected graspability as ground truth.
> > >
> > > **[3.3]** We will update the camera-ready version to feature the new metrics more prominently.

---

### Official Review · Reviewer_Wx5g · 2022-07-23

**Originality:** Good
**Technical Quality:** Very Good
**Clarity Of Presentation:** Very Good
**Impact:** 3

**Recommendation:**

Weak Accept: I recommend accepting the paper, but will not argue for my recommendation if the majority of other reviewers have a different opinion.

**Summary:**

The paper proposes self-supervised learning of dense visual descriptors from unordered RGB images using data augmentation and contrastive losses. This is unlike prior works that require obtaining pixel-level correspondences from camera poses. Training works by sampling pairs of augmentations (affine, perspective, scale, etc) on the same input image, using a ResNet-34 to produce per-pixel descriptors, then using a contrastive loss to align descriptors of corresponding pixels. While this method loses the semantic and category-level information from priors works (e.g. Dense Object Nets), it is much simpler and offers competitive keypoint tracking performance. The authors use the trained descriptors in a downstream bin-picking task, where humans label suction grasp points that are then used to find new grasp points via descriptor matching. The proposed method achieves a 98.9% grasp success rate, higher than the 79.9% achieved by a baseline that does not use these pretrained descriptors.

**Issues:**

Please see strengths/weaknesses section.

**Quality Of The Limitations Section:**

Limitations are addressed clearly

**Reviewer Expertise:**

4: The reviewer is confident but not absolutely certain that the evaluation is correct

**Robotics Focus:**

Sufficient demonstration on hardware

**Strengths And Weaknesses:**

Strengths

The paper is well-written. The method is simple to implement and the experiments are clear. I appreciate the authors’ ablations on the effects of different image augmentation types. Discussion of the failure cases on the model’s relatively poor performance on novel camera perspectives is also appreciated.

Weaknesses

My main complaint about the work is on generalization of the trained descriptor networks on novel objects, and this manifests in a couple different ways. As the authors note, the proposed method works best on known objects. While this method would still be useful (since the network clearly generalizes in other ways, like clutter configuration, camera views, lighting, etc), additional experiments and/or discussions on object-wise generalization would be helpful. In particular, it would be interesting to do the keypoint tracking experiments on completely novel objects, just to see what those numbers are.

For the bin picking experiments, the authors describe the scene as consisting of “ten known objects, but are not contained in the training set.” I’m not sure what this means. One guess is that this means the 10 objects have human-labeled grasps, but they are not in the dataset used to train the descriptor network. If this is true, then does this mean the network does indeed have some generalization capabilities to novel objects?

Lastly, I think the paper would benefit from a quick comparison against training a descriptor network on a much larger but more generic dataset. This would be similar to other works that learn self-supervised image embeddings, except the output is per-pixel instead of per-image. This comparison can help further illustrate the extent of which domain-specific or even instance-specific training data is required.

Perhaps for future work (this would be a stretch for rebuttal, but it’d be impressive), the authors could use an equivariant CNN architecture that is equivariant to translations and rotations for better sample complexity and avoiding certain image augmentation modes. Here’s an example of using equivariant networks in robot manipulation: https://proceedings.mlr.press/v164/wang22j.html

One small comment: Figure 4b should just have 2 colors and 2 line types, instead of 2 colors and 4 line types.


**Summary Of Recommendation:**

I recommend weak accept because while the proposed method could be useful, I can see how generalization may be a contentious point for other reviewers or the community at large.

---

> ### Author Response · Authors · 2022-08-24
> **Answers to Reviewer 2 (Wx5g)**
>
> We thank the reviewer for the useful comments and suggestions. Please see our comments below.
>
> **[2.1] Generalization:** We agree with the reviewer and conducted an additional evaluation on unseen objects to explicitly test and quantify the generalization capabilities of the network. The full results can be found in the rebuttal pdf, Section III.2. For this, we recorded two scenes with 5 completely novel objects. We applied the same evaluation protocol as for the keypoint tracking task of the main section.
>
> We find the median pixel error is minimally increased, while the 75% quantile increased by only 6% percentage points. Similarly, e.g., for the newly added PCK@10 metric the performance is still quite good. Overall, many keypoints are still tracked well, e.g., if only rotations or small perspective distortions occur. However, looking at the 90% and 95% percent quantile we see a 200% increase in pixel error. The criticality of this depends on the application and how keypoints are tracked but can be considered severe.
>
> However, as we train on an unordered set of RGB images, extending the training set is as easy as recording a set of images of scenes containing the new object(s) and re-train the models. We believe this straightforward and efficient approach compensates for the limits of generalizability.
>
> **[2.2] Objects in grasping experiment:** This statement was somewhat misleading. The “not contained in the training set” referred to the specific images, not the objects itself. We used the same objects, but in different configurations in the scene. We improved the description in the paper.
>
> **[2.3] Training on more generic dataset:** Thanks for the excellent observation, we performed the evaluations as you suggested. The results are in the rebuttal pdf Section IV.2.
>
> a) We trained our method not on data from the target domain, but on COCO. In short, for “easy” examples, i.e., small transformations of the input image, the method still performs well. This makes sense since the descriptors learned generic image features which are still preserved over such smaller transformations. However, with stronger perspective transformations it shows that the performance drops considerably. This supports our claim that we need to address this type of transformation explicitly via a diverse, in-distribution dataset during training for the robotic use-case.
>
> b) For completeness, we also compared to one of the most recent keypoint matching algorithms, CATs [A], which is under the top performing methods on various keypoint matching datasets. We took pretrained weights (from dataset PF-Pascal) which result in impressive matching results for general objects. Here, the results were not very good in our target domain, which we account to the following reasons: (a) PF-Pascal has a limited number of classes and the network overfits to those (b) The goal of CAT (and similar approaches) is semantic keypoint matching. In this goal it achieves very impressive results, e.g., the tip of a dog’s nose will be matched to a completely different dog’s nose in a second image. However, the goal of CATs is not the highly accurate matching of geometric points on target objects, which we need for robotic grasping. (See the reported mean pixel error in the original CAT paper).
> We believe this result supports our claim that for the robotics grasping domain we need a dedicated training scheme, towards which we hope to contribute.
>
> [A] Cho, Seokju, et al. "CATs: Cost aggregation transformers for visual correspondence." Advances in Neural Information Processing Systems 34 (2021): 9011-9023.
>
> **[2.4] Equivariant networks:** We agree that equivariant networks with respect to SE(2) or even SE(3) are a very interesting line of research! For use-cases that rely on interaction with the real robot during training, as described in the related approach you mentioned, better sample-efficiency is essential. In our training schema, the data generation pipeline can generate training data with respect to such transformations “for free”. Therefore, we decided to use a vanilla pretrained backbone in our experiments and we achieved good equivariance with respect to SE(3) as a result, as can be seen in the experiments. Still, we have other network architectures as backbones on our list for future work.
>
> We have adapted the related work section accordingly, see rebuttal pdf Section V.
>
> **Minor comments:**
> - We updated figure 4b accordingly

---

> > ### Comment · Reviewer_Wx5g · 2022-08-26
> > **Thank you for the response**
> >
> > I thank the authors for the detailed response and the new experiments, which helped address my initial concerns. The new experiments strengthen the paper and make its claims more convincing. I will be advocating for the paper's acceptance.

---

### Official Review · Reviewer_M14W · 2022-07-30

**Originality:** Fair
**Technical Quality:** Very Good
**Clarity Of Presentation:** Very Good
**Impact:** 3

**Recommendation:**

Strong Accept: I recommend accepting the paper and will argue for my recommendation even if other reviewers hold a different opinion.

**Summary:**

The paper presents a training data augmentation technique to learn local keypoint descriptors which are invariant to certain 2D image transformations. The required setup is simpler than that based on 3D geometric transformations. The authors also present analyses that show performance variations with respect to different 2D transformations and the extent of it. Real world robot experiments are also presented.

**Issues:**

There are no major issues. The presented analysis adds value to the literature. For being more impactful, the paper would need to show how it can combine both 2D and 3D geometric transformation based supervision to achieve state-of-the-art performance.

**Quality Of The Limitations Section:**

Limitations are addressed clearly

**Reviewer Expertise:**

4: The reviewer is confident but not absolutely certain that the evaluation is correct

**Robotics Focus:**

Sufficient demonstration on hardware

**Strengths And Weaknesses:**

**Strengths**

The approach section builds on existing works. The key contribution is in results and analyses which shows how 2D geometric transformations of RGB images can lead to competitive keypoint matching and tracking performance as compared to 3D geometric transformation based training. This is relevant to the field as the approach is simple and leverages the benefits of machine learning where one can use more training data (in a somewhat systematic manner) to achieve certain invariances.
It would have been amazing though to be able to combine the two compares techniques to achieve state-of-the-art performance.

Limitations section is clear and will be helpful in further advances.

Section 4.2 and 4.3 are insightful and clearly show the characteristics of the proposed approach.

Real robot experiments are a plus.

**Weaknesses**

In L114, the authors refer to “synthetic correspondences” in contrast with geometric correspondences. The operations they are performing for synthetic correspondences (described in L125-130) are also geometric as they transform the underlying 2D image pixel grid. The authors need to improve the description of their approach.

L113-114 need clarification: “origin of pixels” and “same origin”

L182 - “unprojects”?


**Summary Of Recommendation:**

The paper clearly conveys its approach and results, which can act as an alternative or complementary way of achieving better results. The novelty lies in results and analyses, not in the approach section, and could still be relevant to the community.

Post-rebuttal:
The authors have done an excellent job in addressing reviewers' concerns. Additional experiments and ablation studies conducted by the authors during the rebuttal are appreciated.

An important point was novelty to which the authors agree that their pipeline uses existing works but the paper mainly argues for simplification of data generation pipeline for training and thus improving performance in the robotics context. As an application, this work seems relevant for the community and is a good validation of 'what one thinks should work does actually work'.

The overall idea and the results are not surprisingly amazing although the execution is great, thus an improved rating.

---

> ### Author Response · Authors · 2022-08-24
> **Answers to Reviewer 1 (M14W)**
>
> We thank the reviewer for the useful comments and suggestions. Please see our comments below.
>
> **[1.1] Defining geometric correspondences:** Thank you for pointing out this possibly confusing part. With synthetic correspondence we refer to obtaining the pixel correspondences via image augmentations, while for geometric correspondence we rely on the 3D geometry of the scene. In the synthetic case we only require a single RGB image, while in the geometric case we need pairs of registered RGBD frames and generate correspondence via multi-view synthesis. We agree with the reviewer that image augmentations also introduce geometric distortions, but we wanted to emphasize with our wording that these are synthetic, as opposed to picking two different views from the 3D reconstruction. We made the description of these definitions clearer in the revision.
>
> **[1.2] Combining 2D and 3D transformations:** Thank you for hinting at this interesting point. Using both geometric correspondences (3D transformations) together with image augmentations (2D transformations) was already studied e.g., by Adrian et. al. [4]. Indeed, they saw an improved performance when using image augmentations together with the 3D transformations for training.
>
> For this paper, however, we propose to train the dense descriptor network explicitly without 3D transformations, relying only on unregistered RGB images. Using 3D transformations for self-supervised training of dense visual descriptors typically requires a complex acquisition of training data. E.g., in Florence et. al. [2] data taking is done using a robot mounted RGBD camera with the need of camera registration. In many applications such a data taking process is not feasible.
>
> A core contribution of our approach is to relax these assumptions and to show that we can train a dense descriptor network in a self-supervised fashion without 3D transformations. It is intentionally not our goal to improve state-of-the-art performance in terms of keypoint tracking accuracy, but to maintain a competitive performance while significantly simplifying the data taking process. Please let us know if you have any further suggestions in this direction.
>
> **Minor Comments:**
> - We improved the description in lines 113-114
> - We clarified “unprojects”

---

> > ### Comment · Reviewer_M14W · 2022-08-26
> > **Thanks for clarifications**
> >
> > The authors' response is satisfactory.

---

### Author Response · Authors · 2022-08-24
**Document Containing Major Changes**

We provide a document containing the major changes and added experiments to the paper as a single pdf. All additional content will be added to the respective section of the final submissions' appendix.

---

### Meta-Review · Area_Chair_q7tH · 2022-08-11

**Recommendation:** Accept (Poster)
**Confidence:** 4

**Metareview:**

Reviewers felt positively that the proposed task of learning visual descriptors is important to robotics.  The proposed approach is simple ,reasonable, and seems to work well.  The results and analysis are insightful, and reviewers also appreciated the real-robot experiments.  The reviewers also felt that the paper is well-written

Reviewers felt that the rebuttal was very convincing and addressed most of the original concerns with the paper. The main remaining reviewer concern is that the proposed method is somewhat incremental, with limited technical novelty compared to prior work. Nonetheless, reviewers felt that the paper is interesting and that members of the robot learning community would benefit from reading it.


**Best Paper Nomination:**

No

---

> ### Author Response · Authors · 2022-08-24
> **Answers to the Area Chair**
>
> We thank all the reviewers and the meta-reviewer for the valuable comments on our paper. We tried to answer all the questions and improved the relevant parts in the paper. We provide a single pdf containing the most important changes and new results, with sections referenced by {Roman numeral}.{Arabic numeral} as a separate post. We reference the individual comments of the reviewers by a number X.Y, for reviewer X and comment number Y.
>
> The new results will all be added to the final submission in the appendix. In the following we would like to address the mentioned weaknesses.
>
> **Limited technical novelty:**
> A recurring comment in the reviews was about the contribution to the current state of the art. We realize now we didn’t highlight our contribution clearly in the paper before. There has been lots of related work on supervised learning of image embeddings [8,10,18] and pixel-wise embeddings [9, 16], where [9] is from the method point of view very close to our approach. And there is much more existing work on keypoint matching in general. However, as we show in our experiments, vanilla embeddings and keypoint matching that work well for general computer vision tasks do not perform well in robotic grasping tasks, where we need to find geometric correspondences over e.g., changing viewpoints or lighting conditions. (And we added another set of experiments in addition to the initial submission to strengthen this point, see IV.2 of our attached rebuttal pdf). To address this problem, Florence et. al. [2] proposed DON, a method to learn embeddings specifically for such applications. The results are promising, but at the cost of high demands on the data generation for the training approach.
>
> In our work, the main contribution is the application and evaluation of state-of-the-art self-supervised learning methods to the robotic domain. This allows us to match the good performance of DONs with a much simpler data generation pipeline. We do believe this is a valuable contribution to the robotics community since it opens further use-cases, e.g., in practically relevant setups such as typical industrial robotic cells with a fixed overhanging camera.
>
> We updated the contribution section in the paper accordingly and hope to clarify this point.
>
> **Better explanation of method:**
> Based on several suggestions of the reviewers we improved the explanation of our method (see e.g., comments by reviewer M14W and aGSf, cf. V).
>
> **Experimental details unclear:**
> We clarified a misunderstanding noticed by several reviewers regarding the objects in the grasping experiment (see e.g., comment 2.2 for reviewer Wx5g).
>
> Furthermore, we improved the description of the experimental details in response to comment 3.2 for reviewer etoZ as well as the motivation of the experiment (see comment 4.2 for aGSf).
>
> **Combination of 2D and 3D transformations:**
> We have answered this point in comment 1.2 for reviewer M14W.
>
> **Limited generalization to new objects:**
> We added an experiment explicitly testing on novel objects to the appendix. We observe some generalization capabilities of our methods as the median pixel error decreases only slightly, however the tails of the pixel error distribution increase significantly. These results indicate that keypoint tracking on unseen objects works on a limited domain. Please refer to our answer to comment 2.1 for further details.
>
> **More extensive experiments:**
> We added several new studies to the paper following the suggestions of the reviewers:
> - Generalization: Testing the proposed method on unseen objects, see comment 2.1 reviewer Wx5g. [III.2]
> - Training on more generic dataset, see comment 2.3 reviewer Wx5g. [IV.2]
> - Comparing against a SoTA generic keypoint matching algorithm, comment 2.3 reviewer Wx5g . [IV.2]
> - Ablation study evaluating probability of augmentations, see comment 3.4, reviewer etoZ. [IV.1]